# The Closeness of In-Context Learning and Weight Shifting for Softmax Regression

**Shuai Li**
Shanghai Jiao Tong University
shuaili8@sjtu.edu.cn

**Zhao Song**
Simons Institute for the Theory of Computing, UC Berkeley
magic.linuxkde@gmail.com

**Yu Xia**
University of California, San Diego
yux078@ucsd.edu

**Tong Yu**
Adobe Research
tyu@adobe.com

**Tianyi Zhou**
University of Southern California
tzhou029@usc.edu

## Abstract

Large language models (LLMs) are known for their exceptional performance in natural language processing, making them highly effective in many human life-related tasks. The attention mechanism in the Transformer architecture is a critical component of LLMs, as it allows the model to selectively focus on specific input parts. The softmax unit, which is a key part of the attention mechanism, normalizes the attention scores. Hence, the performance of LLMs in various NLP tasks depends significantly on the crucial role played by the attention mechanism with the softmax unit.

In-context learning is one of the celebrated abilities of recent LLMs. Without further parameter updates, Transformers can learn to predict based on few in-context examples. However, the reason why Transformers becomes in-context learners is not well understood. Recently, in-context learning has been studied from a mathematical perspective with simplified linear self-attention without softmax unit. Based on a linear regression formulation $\min_x \|Ax - b\|_2$, existing works show linear Transformers' capability of learning linear functions in context. The capability of Transformers with softmax unit approaching full Transformers, however, remains unexplored.

In this work, we study the in-context learning based on a softmax regression formulation $\min_x \|\langle \exp(Ax), \mathbf{1}_n \rangle^{-1} \exp(Ax) - b\|_2$. We show the upper bounds of the data transformations induced by a single self-attention layer with softmax unit and by gradient-descent on a $\ell_2$ regression loss for softmax prediction function. Our theoretical results imply that when training self-attention-only Transformers for fundamental regression tasks, the models learned by gradient-descent and Transformers show great similarity.

## 1 Introduction

In recent years, there has been a significant increase in research and development in the field of Artificial Intelligence, with large language models (LLMs) emerging as an effective way to tackle complex tasks. Transformers have achieved state-of-the-art results in various NLP tasks, such as machine translation [PCR19, GHG$^+$20] and text generation [LSX$^+$22]. As a result, they have become the preferred architecture for NLP, where BERT [DCLT18], GPT-3 [BMR$^+$20], PaLM [CND$^+$22] were proposed. They have demonstrated remarkable learning and reasoning capabilities and have proven to be more efficient than traditional models when processing natural language.

38th Conference on Neural Information Processing Systems (NeurIPS 2024).

Additionally, LLMs can be fine-tuned for multiple purposes without requiring a new build from scratch, making them a versatile tool for AI applications. Moreover, recent studies on the in-context learning abilities of LLMs have demonstrated that even without further fine-tuning, LLMs can generalize to new tasks with only a few demonstration examples in the prompt. To understand how LLMs become in-context learners, recent works have studied the problem and provided mathematical explanations from the Transformer architecture perspective, showing a simplified linear self-attention layer of Transformer can learn linear functions similarly as a step of gradient descent [ONR$^+$22, ASA$^+$22, GTLV22, CLL$^+$24]. While such linear approximation of full Transformers is overly simplistic, studies on more complex Transformer architecture are needed to further explain the in-context learning phenomenon.

Transformers have a specific type of sequence-to-sequence neural network architecture. They utilize the attention mechanism [VSP$^+$17, RNS$^+$18, DCLT18, BMR$^+$20] that allows them to capture long-range dependencies and context from input data effectively. The core of the attention mechanism is the attention matrix which is comprised of rows and columns, corresponding to individual words or "tokens". The attention matrix represents the relationships within the given text. It measures the importance of each token in a sequence as it relates to the desired output. During the training process, the attention matrix is learned and optimized to improve the accuracy of the model's predictions. Through the attention mechanism, each input token is evaluated based on its relevance to the desired output by assigning a token score. This score is determined by a similarity function that compares the current output state with input states.

Theoretically, the attention matrix is comprised of the query matrix $Q \in \mathbb{R}^{n \times d}$, the key matrix $K \in \mathbb{R}^{n \times d}$ and the value matrix $V \in \mathbb{R}^{n \times d}$. Following [ZHDK23, AS23, BSZ24, AS24b, AS24c, AS24a], the computation of the normalized attention function is defined as $D^{-1} \exp(QK^\top)V$. Following the transformer literature, we apply $\exp$ to a matrix entry-wise way. Here $D \in \mathbb{R}^{n \times n}$ is diagonal matrix that defined as $D = \mathrm{diag}(\exp(QK^\top)\mathbf{1}_n)$. Intuitively, $D$ denotes the softmax normalization matrix. A more general computation formulation can be written as

$$\underbrace{D^{-1}}_{n \times n \text{ diagonal matrix}} \underbrace{\exp(XQK^\top X^\top)}_{n \times n} \underbrace{X}_{n \times d} \underbrace{V}_{d \times d},$$

where

$$D := \mathrm{diag}(\exp(XQK^\top X^\top)\mathbf{1}_n).$$

In the above setting, we treat $Q, K, V \in \mathbb{R}^{d \times d}$ as weights and $X$ is the input sentence data that has length $n$ and each word embedding size is $d$. In the remaining of the part, we will switch $X$ to notation $A$ and use $A$ to denote sentence. Mathematically, the attention computation problem can be formulated as a regression problem in the following sense

**Definition 1.1.** *We consider the following problem*

$$\min_{X \in \mathbb{R}^{d \times d}} \|D^{-1} \exp(AXA^\top) - B\|_F$$

*where $A \in \mathbb{R}^{n \times d}$ can be treated as a length-$n$ document and each word has length-$d$ embedding size. Here $D = \mathrm{diag}(AXA^\top \mathbf{1}_n)$. For any given $A \in \mathbb{R}^{n \times d}$ and $B \in \mathbb{R}^{n \times n}$, the goal is to find some weight $X$ to optimize the above objective function.*

In contrast to the formulation in [ZHDK23, AS23, BSZ24], the parameter $X$ in Definition 1.1 is equivalent to the $QK^\top \in \mathbb{R}^{d \times d}$ in the generalized version of [ZHDK23, AS23, BSZ24] (e.g. replacing $Q \in \mathbb{R}^{n \times d}$ by $XQ$ where $X \in \mathbb{R}^{n \times d}$ and $Q \in \mathbb{R}^{d \times d}$. Similarly for $K$ and $V$. In such scenario, $X$ can be viewed as a matrix representation of a length-$n$ sentence.).

A number of work [ASA$^+$22, GTLV22, ONR$^+$22] study the in-context learning from mathematical perspective in a much simplified setting than Definition 1.1, which is linear regression formulation as in Definition 1.2. They show linear Transformer without softmax unit in its attention layer can mimic the ability of gradient descent in learning linear functions in context. While the softmax unit plays an important role in attention computations of full Transformers, their simplification of the softmax unit leaves a gap in explaining LLMs' in-context learning abilities.

**Definition 1.2.** *Given a matrix $A \in \mathbb{R}^{n \times d}$ and $b \in \mathbb{R}^n$, the goal is to solve*

$$\min_x \|Ax - b\|_2$$

Several theoretical transformer work have studied either exponential regression [GMS23, LSZ23] or softmax regression problem [DLS23, LLSS24a]. In this work, to take a step forward to understand the softmax unit in the attention scheme in LLMs. We consider the following softmax regression and study the in-context learning phenomena and its closeness to gradient descent from the data transformation perspective.

**Definition 1.3** (Softmax Regression). *Given a $A \in \mathbb{R}^{n \times d}$ and a vector $b \in \mathbb{R}^n$, the goal is to solve*

$$\min_{x \in \mathbb{R}^d} \|\langle \exp(Ax), \mathbf{1}_n \rangle^{-1} \exp(Ax) - b\|_2$$

We remark that the Definition 1.3 of Softmax Regression is a formulation in between Definition 1.2 and Definition 1.1.

We state our major result as follows:

**Theorem 1.4** (Bounded shift for in-context learning, informal version of the combination of Theorem 4.2 and Theorem 4.3). *If the following conditions hold: Let $A \in \mathbb{R}^{n \times d}$. Let $b \in \mathbb{R}^n$. $\|A\| \leq R$. Let $\|x\|_2 \leq R$. $\|A(x_{t+1} - x_t)\|_\infty < 0.01$. $\|(A_{t+1} - A_t)x\|_\infty < 0.01$. Let $R \geq 4$. Let $M := n^{1.5} \exp(10R^2)$. We consider the softmax regression (Definition 1.3) problem*

$$\min_x \|\langle \exp(Ax), \mathbf{1}_n \rangle^{-1} \exp(Ax) - b\|_2.$$

- **Part 1.** *If we move the $x_t$ to $x_{t+1}$, then we're solving a new softmax regression with $\min_x \|\langle \exp(Ax), \mathbf{1}_n \rangle^{-1} \exp(Ax) - \widetilde{b}\|_2$ where $\|\widetilde{b} - b\|_2 \leq M \cdot \|x_{t+1} - x_t\|_2$*

- **Part 2.** *If we move the $A_t$ to $A_{t+1}$, then we're solving a new softmax regression with $\min_x \|\langle \exp(Ax), \mathbf{1}_n \rangle^{-1} \exp(Ax) - \widehat{b}\|_2$ where $\|\widehat{b} - b\|_2 \leq M \cdot \|A_{t+1} - A_t\|$*

Recall that $A \in \mathbb{R}^{n \times d}$ denotes a length-$n$ document and each word has the length-$d$ embedding size and $x$ denotes the simplified version of $QK^\top$. One-step gradient descent can be treated as an update to the model's weight $x$. Thus, part 1 of our result (Theorem 1.4) implies that the data transformation of $b$ induced by gradient-descent on the $\ell_2$ regression loss is bounded by $M \cdot \|x_{t+1} - x_t\|_2$.

According to [ONR+22], to do in-context learning, a self-attention layer update can be treated as an update to the tokenized document $A$. For detailed derivation, please refer to [ONR+22]. Thus, part 2 of our result (Theorem 1.4) implies that the data transformation of $b$ induced by a single self-attention layer is bounded by $M \cdot \|A_{t+1} - A_t\|$.

We remark that the data transformation of $b$ induced by 1) a single self-attention layer and by 2) gradient-descent on the $\ell_2$ regression loss are both bounded. The bounded transformation of $b$ implies that when training self-attention-only Transformers for fundamental regression tasks, the models learned by gradient-descent and Transformers show great similarity.

**Roadmap.** In Section 2, we give some preliminaries. In Section 3, we compute the gradient of the loss function with softmax function with respect to $x$. Those functions include $\alpha(x)^{-1}$, $\alpha(x)$ and $f(x)$. In Section 4, we give our formal theoretical results, validated by numerical experiments presented in Section 5. In Section 6, we conclude our paper.

## 2 Preliminary

In Section 2.1, we introduce the notations used in this paper. In Section 2.2, we give some facts about the basic algebra. In Section 2.3, we propose the lower bound on $\langle \exp(Ax), \mathbf{1}_n \rangle$.

### 2.1 Notations

For a positive integer $n$, we use $[n]$ to denote $\{1, 2, \cdots, n\}$, for any positive integer $n$. We use $\mathbb{E}[\cdot]$ to denote expectation. We use $\Pr[\cdot]$ to denote probability. We use $\mathbf{1}_n$ to denote the vector where all entries are one. We use $\mathbf{0}_0$ to denote the vector where all entries are zero. The identity matrix of size $n \times n$ is represented by $I_n$ for a positive integer $n$. The symbol $\mathbb{R}$ refers to real numbers and $\mathbb{R}_{\geq 0}$ represents non-negative real numbers. For any vector $x \in \mathbb{R}^n$, $\exp(x) \in \mathbb{R}^n$ denotes a vector where the $i$-th entry is $\exp(x_i)$ and $\|x\|_2$ represents its $\ell_2$ norm, that is, $\|x\|_2 := (\sum_{i=1}^n x_i^2)^{1/2}$. We use

$\|x\|_\infty$ to denote $\max_{i \in [n]} |x_i|$. For any vector $x \in \mathbb{R}^n$ and vector $y \in \mathbb{R}^d$, we use $\langle x, y \rangle$ to denote the inner product of vector $x$ and $y$. The notation $B_i$ is used to indicate the $i$-th row of matrix $B$. If $a$ and $b$ are two column vectors in $\mathbb{R}^n$, then $a \circ b$ denotes a column vector where $(a \circ b)_i = a_i b_i$. For a square and full rank matrix $B$, we use $B^{-1}$ to denote the true inverse of $B$.

## 2.2 Basic Algebras

**Fact 2.1.** *For vectors $x, y \in \mathbb{R}^n$, we have*

- $\|x \circ y\|_2 \leq \|x\|_\infty \cdot \|y\|_2$
- $\|x\|_\infty \leq \|x\|_2 \leq \sqrt{n}\|x\|_\infty$
- $\|\exp(x)\|_\infty \leq \exp(\|x\|_2)$
- *For any $\|x - y\|_\infty \leq 0.01$, we have $\|\exp(x) - \exp(y)\|_2 \leq \|\exp(x)\|_2 \cdot 2\|x - y\|_\infty$*

**Fact 2.2.** *For matrices $X, Y$, we have*

- $\|X^\top\| = \|X\|$
- $\|X\| \geq \|Y\| - \|X - Y\|$
- $\|X + Y\| \leq \|X\| + \|Y\|$
- $\|X \cdot Y\| \leq \|X\| \cdot \|Y\|$
- *If $X \preceq \alpha \cdot Y$, then $\|X\| \leq \alpha \cdot \|Y\|$*

## 2.3 Lower bound on $\beta$

**Lemma 2.3.** *If the following conditions holds*

- $\|A\| \leq R$
- $\|x\|_2 \leq R$
- *Let $\beta$ be lower bound on $\langle \exp(Ax), \mathbf{1}_n \rangle$*

*Then we have*

$$\beta \geq \exp(-R^2)$$

*Proof.* We have

$$
\begin{aligned}
\langle \exp(Ax), \mathbf{1}_n \rangle &= \sum_{i=1}^n \exp((Ax)_i) \\
&\geq \min_{i \in [n]} \exp((Ax)_i) \\
&\geq \min_{i \in [n]} \exp(-|(Ax)_i|) \\
&= \exp(-\max_{i \in [n]} |(Ax)_i|) \\
&= \exp(-\|Ax\|_\infty) \\
&\geq \exp(-\|Ax\|_2) \\
&\geq \exp(-R^2)
\end{aligned}
$$

the 1st step follows from simple algebra, the 2nd step comes from simple algebra, the 3rd step follows from the fact that $\exp(x) \geq \exp(-|x|)$, the 4th step follows from the fact that $\exp(-x)$ is monotonically decreasing, the 5th step comes from definition of $\ell_\infty$ norm, the 6th step follows from Fact 2.1, the 7th step follows from the assumption on $A$ and $x$. $\qquad \square$

# 3  Softmax Function with Respect to $x$

In Section 3.1, we give the definitions used in the computation. In Section 3.2, we compute the gradient of the loss function with softmax function with respect to $x$. Those functions includes $\alpha(x)^{-1}$, $\alpha(x)$ and $f(x)$.

## 3.1  Definitions

We define function softmax $f$ as follows

**Definition 3.1** (Function $f$, Definition 5.1 in [DLS23])**.** *Given a matrix $A \in \mathbb{R}^{n \times d}$. Let $\mathbf{1}_n$ denote a length-$n$ vector that all entries are ones. We define prediction function $f : \mathbb{R}^d \to \mathbb{R}^n$ as follows*

$$f(x) := \langle \exp(Ax), \mathbf{1}_n \rangle^{-1} \cdot \exp(Ax).$$

**Definition 3.2** (Loss function $L_{\exp}$, Definition 5.3 in [DLS23])**.** *Given a matrix $A \in \mathbb{R}^{n \times d}$ and a vector $b \in \mathbb{R}^n$. We define loss function $L_{\exp} : \mathbb{R}^d \to \mathbb{R}$ as follows*

$$L_{\exp}(x) := 0.5 \cdot \| \langle \exp(Ax), \mathbf{1}_n \rangle^{-1} \exp(Ax) - b \|_2^2.$$

For convenient, we define two helpful notations $\alpha$ and $c$

**Definition 3.3** (Normalized coefficients, Definition 5.4 in [DLS23])**.** *We define $\alpha : \mathbb{R}^d \to \mathbb{R}$ as follows*

$$\alpha(x) := \langle \exp(Ax), \mathbf{1}_n \rangle.$$

*Then, we can rewrite $f(x)$ (see Definition 3.1) and $L_{\exp}(x)$ (see Definition 3.2) as follows*

- $f(x) = \alpha(x)^{-1} \cdot \exp(Ax).$

- $L_{\exp}(x) = 0.5 \cdot \| \alpha(x)^{-1} \cdot \exp(Ax) - b \|_2^2.$

- $L_{\exp}(x) = 0.5 \cdot \| f(x) - b \|_2^2.$

**Definition 3.4** (Definition 5.5 in [DLS23])**.** *We define function $c : \mathbb{R}^d \in \mathbb{R}^n$ as follows*

$$c(x) := f(x) - b.$$

*Then we can rewrite $L_{\exp}(x)$ (see Definition 3.2) as follows*

- $L_{\exp}(x) = 0.5 \cdot \| c(x) \|_2^2.$

## 3.2  Gradient Computations

We state a lemma from previous work,

**Lemma 3.5** (Gradient, Lemma 5.6 in [DLS23])**.** *If the following conditions hold*

- *Given matrix $A \in \mathbb{R}^{n \times d}$ and a vector $b \in \mathbb{R}^n$.*

- *Let $\alpha(x)$ be defined in Definition 3.3.*

- *Let $f(x)$ be defined in Definition 3.1.*

- *Let $c(x)$ be defined in Definition 3.4.*

- *Let $L_{\exp}(x)$ be defined in Definition 3.2.*

*For each $i \in [d]$, we have*

- *Part 1.*

$$\frac{\mathrm{d} \exp(Ax)}{\mathrm{d} x_i} = \exp(Ax) \circ A_{*,i}$$

- *Part 2.*

$$\frac{\mathrm{d}\langle \exp(Ax), \mathbf{1}_n\rangle}{\mathrm{d}x_i} = \langle \exp(Ax), A_{*,i}\rangle$$

- *Part 3.*

$$\frac{\mathrm{d}\alpha(x)^{-1}}{\mathrm{d}x_i} = -\alpha(x)^{-1} \cdot \langle f(x), A_{*,i}\rangle$$

- *Part 4.*

$$\frac{\mathrm{d}f(x)}{\mathrm{d}x_i} = \frac{\mathrm{d}c(x)}{\mathrm{d}x_i} = -\langle f(x), A_{*,i}\rangle \cdot f(x) + f(x) \circ A_{*,i}$$

- *Part 5.*

$$\frac{\mathrm{d}L_{\exp}(x)}{\mathrm{d}x_i} = \underbrace{A_{*,i}^\top}_{1\times n} \cdot \Big( \underbrace{f(x)}_{n\times 1} \underbrace{\langle c(x), f(x)\rangle}_{\text{scalar}} + \underbrace{\operatorname{diag}(f(x))}_{n\times n} \underbrace{c(x)}_{n\times 1} \Big)$$

# 4   Main Results

In Section 4.1, we show the lipschitz bound of function $f$. In Section 4.2, we show our upper bound result of $\delta_b$ with respect to $x$. In Section 4.3, we show our upper bound result of $\delta_b$ with respect to $A$.

## 4.1   Lipschitz Bound

To bound the shift of $b$, we first show the Lipschitz property for the basic functions:

- $\| \exp(Ax) - \exp(Ay)\|_2 \leq 2\sqrt{n}R\exp(R^2) \cdot \|x - y\|_2$
- $|\alpha(x) - \alpha(y)| \leq \| \exp(Ax) - \exp(Ay)\|_2 \cdot \sqrt{n}$
- $|\alpha(x)^{-1} - \alpha(y)^{-1}| \leq \beta^{-2} \cdot |\alpha(x) - \alpha(y)|$

We can show that

**Lemma 4.1.** *If the following conditions hold*

- *Let $\beta \in (0, 1)$.*

- *Let $\delta_{b,1} \in \mathbb{R}^n$ be defined as Definition B.3.*

- *Let $\delta_{b,2} \in \mathbb{R}^n$ be defined as Definition B.3.*

- *Let $\delta_b = \delta_{b,1} + \delta_{b,2}$.*

- *Let $R \geq 4$.*

*We have*

- *Part 1.*

$$\|\delta_{b,1}\|_2 \leq 2\beta^{-2}n^{1.5}\exp(2R^2) \cdot \|x_{t+1} - x_t\|_2$$

- *Part 2.*

$$\|\delta_{b,2}\|_2 \leq 2\beta^{-1}\sqrt{n}R\exp(R^2) \cdot \|x_{t+1} - x_t\|_2$$

- *Part 3.*

$$\| \underbrace{f(x_{t+1}) - f(x_t)}_{\delta_b} \|_2 \leq 4\beta^{-2}n^{1.5}R\exp(2R^2) \cdot \|x_{t+1} - x_t\|_2$$

*Proof.* **Proof of Part 1.** We have

$$
\begin{aligned}
\|\delta_{b,1}\|_2 &\leq |\alpha(x_{t+1})^{-1} - \alpha(x_t)^{-1}| \cdot \|\exp(Ax_{t+1})\|_2 \\
&\leq |\alpha(x_{t+1})^{-1} - \alpha(x_t)^{-1}| \cdot \sqrt{n} \cdot \exp(R^2) \\
&\leq \beta^{-2} \cdot |\alpha(x_{t+1}) - \alpha(x_t)| \cdot \sqrt{n} \cdot \exp(R^2) \\
&\leq \beta^{-2} \cdot \sqrt{n} \cdot \|\exp(Ax_{t+1}) - \exp(Ax_t)\|_2 \cdot \sqrt{n} \cdot \exp(R^2) \\
&\leq \beta^{-2} \cdot \sqrt{n} \cdot 2\sqrt{n}R\exp(R^2)\|x_{t+1} - x_t\|_2 \cdot \sqrt{n} \cdot \exp(R^2) \\
&= 2\beta^{-2}n^{1.5}R\exp(2R^2) \cdot \|x_{t+1} - x_t\|_2
\end{aligned}
$$

where the first step follows from definition, the second step follows from assumption on $A$ and $x$, the third step follows Lemma B.7, the forth step follows from Lemma B.6, the fifth step follows from Lemma B.5.

**Proof of Part 2.**

We have

$$
\begin{aligned}
\|\delta_{b,2}\|_2 &\leq |\alpha(x_{t+1})^{-1}| \cdot \|\exp(Ax_{t+1}) - \exp(Ax_t)\|_2 \\
&\leq \beta^{-1} \cdot \|\exp(Ax_{t+1}) - \exp(Ax_t)\|_2 \\
&\leq \beta^{-1} \cdot 2\sqrt{n}R\exp(2R^2) \cdot \|x_{t+1} - x_t\|_2
\end{aligned}
$$

where the first step follows from definition, the 2nd step comes from Lemma B.5.

**Proof of Part 3.**

We have

$$
\begin{aligned}
\|\delta_b\|_2 &= \|\delta_{b,1} + \delta_{b,2}\|_2 \\
&\leq \|\delta_{b,1}\|_2 + \|\delta_{b,2}\|_2 \\
&\leq 2\beta^{-2}n^{1.5}R\exp(2R^2) \cdot \|x_{t+1} - x_t\|_2 + 2\beta^{-1}n^{0.5}R\exp(2R^2) \cdot \|x_{t+1} - x_t\|_2 \\
&\leq 2\beta^{-2}n^{1.5}R\exp(2R^2) \cdot \|x_{t+1} - x_t\|_2 + 2\beta^{-2}n^{1.5}R\exp(2R^2) \cdot \|x_{t+1} - x_t\|_2 \\
&\leq 4\beta^{-2}n^{1.5}R\exp(2R^2) \cdot \|x_{t+1} - x_t\|_2
\end{aligned}
$$

where the 1st step follows from the definition of $\delta_b$, the 2nd step follows from triangle inequality, the 3rd step follows from the results in Part 1 and Part 2, the 4th step follows from the fact that $n \geq 1$ and $\beta^{-1} \geq 1$, the 5th step follows from simple algebra. $\qquad\square$

Similarly, we can show the Lipschitz property of function $f$ with respect to $A$ as the following

$$
\begin{aligned}
&\|f(A_{t+1}) - f(A_t)\|_2 \\
&\leq 4\beta^{-2}n^{1.5}R\exp(2R^2) \cdot \|A_{t+1} - A_t\|_2
\end{aligned}
$$

Due to space limitation, we defer formal lemma and proof to D.2.

## 4.2 Shifting Weight Parameter $x$

**Theorem 4.2** (Bounded shift for shifting the weight parameter, formal of Theorem 1.4)**.** *If the following conditions hold*

- *Let $A \in \mathbb{R}^{n \times d}$*

- *$\|A\| \leq R$*

- *$\|A(x_{t+1} - x_t)\|_\infty < 0.01$*

- *Let $R \geq 4$*

- *Let $M := n^{1.5}\exp(10R^2)$.*

*We consider the softmax regression problem*

$$\min_x \|\langle \exp(Ax), \mathbf{1}_n \rangle^{-1} \exp(Ax) - b\|_2$$

*If we move the $x_t$ to $x_{t+1}$, then we're solving a new softmax regression problem with*

$$\min_x \|\langle \exp(Ax), \mathbf{1}_n \rangle^{-1} \exp(Ax) - \widetilde{b}\|_2$$

*where*

$$\|\widetilde{b} - b\|_2 \le M \cdot \|x_{t+1} - x_t\|_2$$

*Proof.* We have

$$
\begin{aligned}
\|\widetilde{b} - b\|_2 &\le 4\beta^{-2} n^{1.5} R \exp(2R^2) \cdot \|x_{t+1} - x_t\|_2 \\
&\le 4 n^{1.5} R \exp(2R^2) \exp(2R^2) \cdot \|x_{t+1} - x_t\|_2 \\
&\le n^{1.5} (4R) \exp(4R^2) \cdot \|x_{t+1} - x_t\|_2 \\
&\le n^{1.5} \exp(6R^2) \exp(4R^2) \cdot \|x_{t+1} - x_t\|_2 \\
&\le n^{1.5} \exp(10R^2) \cdot \|x_{t+1} - x_t\|_2 \\
&\le M \cdot \|x_{t+1} - x_t\|_2
\end{aligned}
$$

where the 1st step follows from Lemma 4.1, the 2nd step comes from Lemma 2.3, the 3rd step comes from simple algebra, the 4th step follows from simple algebra, the 5th step follows from simple algebra and the 6th step follows from the definition of $M$. □

### 4.3 Shifting Sentence Data $A$

**Theorem 4.3** (Bounded shift for in-context learning, formal of Theorem 1.4). *If the following conditions hold*

- *Let $A \in \mathbb{R}^{n \times d}$*

- *$\|A\| \le R$*

- *$\|(A_{t+1} - A_t)x\|_\infty < 0.01$*

- *Let $R \ge 4$*

- *Let $M := n^{1.5} \exp(10R^2)$.*

*We consider the softmax regression problem*

$$\min_x \|\langle \exp(Ax), \mathbf{1}_n \rangle^{-1} \exp(Ax) - b\|_2$$

*If we move the $A_t$ to $A_{t+1}$ then we're solving a new softmax regression problem with*

$$\min_x \|\langle \exp(Ax), \mathbf{1}_n \rangle^{-1} \exp(Ax) - \widetilde{b}\|_2$$

*where*

$$\|\widetilde{b} - b\|_2 \le M \cdot \|A_{t+1} - A_t\|.$$

*Proof.* We have

$$
\begin{aligned}
\|\widetilde{b} - b\|_2 &\le 4\beta^{-2} n^{1.5} R \exp(2R^2) \cdot \|A_{t+1} - A_t\| \\
&\le 4 n^{1.5} R \exp(2R^2) \exp(2R^2) \cdot \|A_{t+1} - A_t\| \\
&\le n^{1.5} (4R) \exp(4R^2) \cdot \|A_{t+1} - A_t\| \\
&\le n^{1.5} \exp(6R^2) \exp(4R^2) \cdot \|A_{t+1} - A_t\| \\
&\le n^{1.5} \exp(10R^2) \cdot \|A_{t+1} - A_t\| \\
&\le M \cdot \|A_{t+1} - A_t\|
\end{aligned}
$$

where the 1st step follows from Lemma D.5, the 2nd step follows from Lemma 2.3, the 3rd step follows from simple algebra, the 4th step comes from simple algebra, the 5th step comes from simple algebra and the 6th step follows from the definition of $M$. □

# 5 Numerical Experiments

In this section, we present our numerical experiments to validate our theoretical results that when training self-attention-only Transformers for softmax regression tasks, the models learned by gradient-descent and Transformers show great similarity.

## 5.1 Experiments Setup

According to Definition 1.3, we construct the synthetic softmax regression tasks consists of randomly sampled length-$n$ documents $A \in \mathbb{R}^{n \times d}$ where each word has the $d$-dimensional embedding and targets $b \in \mathbb{R}^n$. Each document is generated from a unique random seed. In our experiments, we choose a set of different document length $n$ and a set of different embedding size $d$.

Following [ONR+22], we compare the following two models in our experiment

- a trained single self-attention (SA) layer with softmax unit approximating full Transformers.
- a softmax regression model trained with one-step gradient descent (GD).

The training objective for both models is defined as in Definition 1.3. For the single self-attention layer with a softmax unit, we choose the learning rate $\eta_{\mathrm{SA}} = 0.005$. For the softmax regression model, we determine the optimal learning rate $\eta_{\mathrm{GD}}$ by minimizing the $\ell_2$ regression loss over a training set of $10^3$ tasks through line search.

To compare the trained single self-attention layer with a softmax unit and the softmax regression model trained with one-step gradient descent, we sample $10^3$ tasks and record the losses of two models. In addition, we follow [ONR+22] to record

- **Pred Diff**: the predictions difference measured with the $\ell_2$ norm:

$$\|\widehat{y}_{\mathrm{SA}}(A) - \widehat{y}_{\mathrm{GD}}(x)\|_2$$

  where $\widehat{y}_{\mathrm{SA}}(A)$ corresponds to the $\widetilde{b}$ in Theorem 4.2, and $\widehat{y}_{\mathrm{GD}}(x)$ corresponds to the $\widetilde{b}$ in Theorem 4.3.
- **Model Cos**: the cosine similarity between the sensitivities of two models:

$$\mathtt{CosSim}\left(\frac{\partial \widehat{y}_{\mathrm{GD}}(x)}{\partial x}, \frac{\partial \widehat{y}_{\mathrm{SA}}(A)}{\partial A}\right)$$

- **Model Diff**: the model sensitivity difference measured with the $\ell_2$ norm:

$$\|\frac{\partial \widehat{y}_{\mathrm{GD}}(x)}{\partial x} - \frac{\partial \widehat{y}_{\mathrm{SA}}(A)}{\partial A}\|_2$$

All experiments run on a single NVIDIA RTX2080Ti GPU with 10 independent repetitions.

## 5.2 Different Document Lengths

For synthetic softmax regression tasks of document length $n \in \{200, 1000\}$ and word embedding size $d = 20$, the comparison results between a trained single self-attention layer and one-step gradient descent are shown in Figure 1 and Figure 2. Due to space limitation, we present more results with different document length $n \in \{25, 50, 100, 200, 400, 1000\}$ in Appendix E.

We compare two models' losses over training steps of Transformers in Figure 1a and Figure 2a. In Figure 1b and Figure 2b, we show the differences and similarities of two models over the training steps. From the results, we find identical performances of the two models measured in losses. We also observe considerable alignment of the two models across tasks of different document lengths, indicated by decreasing prediction and model difference and increasing cosine similarity between models. Besides, comparing results with different $n$, we observe that with larger document length, which is common in practical NLP tasks, more training steps are required for Transformers to exhibit such similarities. This shows the crucial role of pretraining stage of Transformers for their in-context learning ability.

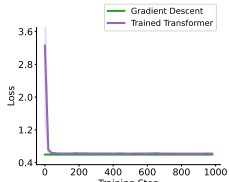 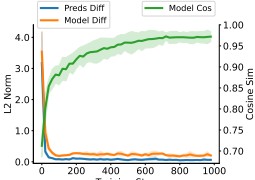 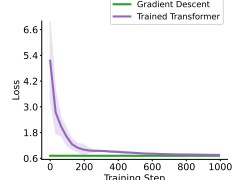 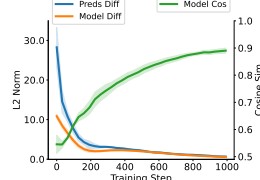

(a) Losses over training steps of Transformer

(b) Difference and similarity over training steps

(a) Losses over training steps of Transformer

(b) Difference and similarity over training steps

Figure 1: Comparison on softmax regression tasks of document length $n = 200$.

Figure 2: Comparison on softmax regression tasks of document length $n = 1000$.

## 5.3 Different Word Embedding Sizes

We also compare trained single self-attention layer and one-step gradient descent on synthetic softmax regression tasks of different word embedding sizes and document length $n = 200$. Similarly, we measure two models' losses and similarities over training steps on each set of tasks. Due to space limitation, we follow [ONR+22] to show in Figure 3 the loss comparisons at the end of training over different embedding size $d \in \{5, 10, 20, 35, 50\}$. The complete loss curves and measurements of model difference and similarity are presented in Appendix E.

From the results, we again observe similar performances and close alignment of the two models with different word embedding sizes.

To summarize, our numerical results validate our theoretical results in Section 4, showing that when training self-attention-only Transformers for softmax regression tasks, the models learned by gradient-descent and Transformers show great similarity. Note that due to the non-linearity of softmax regression, it is not expected for models to match exactly as implied in our theoretical results in Section 4, which is also observed in our numerical findings.

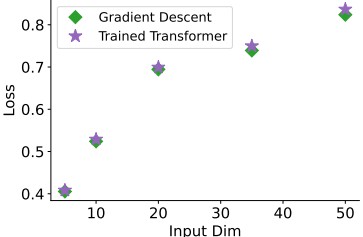

Figure 3: Loss comparisons with different word embedding sizes $d$.

## 6 Conclusion

The attention mechanism that incorporates the softmax unit is a crucial aspect of Large Language Models (LLMs) and significantly contributes to their extraordinary performance in various Natural Language Processing (NLP) tasks. The ability to learn in-context is highly valued in recent LLMs, and comprehending this concept is vital when querying LLMs. In this study, taking a step further from prior works' studies on linear Transformer's ability of learning linear functions, we examined the in-context learning process from a softmax regression perspective of Transformer's attention mechanism. We established the bound on the data transformations brought about by a single self-attention layer with softmax unit and gradient descent on an L2 regression loss. Our findings suggest that the update acquired through gradient descent and in-context learning are highly similar when training self-attention-only Transformers for softmax regression tasks, which is also validated through our preliminary experimental results. These results offer insights into the theoretical underpinnings of in-context learning in Transformers and can aid in improving the understanding and performance of LLMs in various NLP tasks.

## Acknowledgments

The authors would like to thank Jerry Yao-Chieh Hu, Zhenmei Shi, Lichen Zhang and Yufa Zhou for helping preparing for camera-ready version of this paper. For more information related to the paper and adjacent topics, see https://www.youtube.com/@zhaosong2031 and https://space.bilibili.com/3546587376650961.

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

# Appendix

**Roadmap.** In Section A, we introduce some related works. In Section B, we compute the Lipschitz with respect to $x$. In Section C, we give some definitions related to the softmax function of $A$. In Section D, we compute the Lipschitz with respect to $A$. In Section E, we show our complete numerical experiments that support our theoretical results.

# A  Related Work

## A.1  In-Context Learning

[ASA$^+$22] indicate that Transformer-based in-context learners are able to perform traditional learning algorithms implicitly. This is achieved by encoding smaller models within their internal activations. These smaller models are updated by the given context. They theoretically investigate the learning algorithms that Transformer decoders can implement. They demonstrate that Transformers need only a limited number of layers and hidden units to implement various linear regression algorithms. For $d$-dimensional regression problems, a $O(d)$-hidden-size Transformer can perform a single step of gradient descent. They also demonstrate its ability to update a ridge regression problem. The study reveals that Transformers theoretically have the ability to perform multiple linear regression algorithms.

[GTLV22] concentrate on training Transformer to learn certain functions, under in-context conditions. The goal is to have a more comprehensive understanding of in-context learning and determine if Transformers can learn the majority of functions within a given class after training. They found that in-context learning is possible even when there is a distribution shift between training and inference data or between in-context examples and query inputs. In addition, they find out that Transformers can learn more complex function classes such as sparse linear functions, two-layer neural networks, and decision trees. These trained Transformers have comparable performance to task-specific learning algorithms.

[ONR$^+$22] demonstrate the similarity between the training process of Transformers in in-context tasks and some meta-learning formulations based on gradient descent. During training Transformers for auto-regressive tasks, the implementation of in-context learning in the Transformer forward pass is carried out through gradient-based optimization of an implicit auto-regressive inner loss that is constructed from the in-context data.

Formally speaking, they consider the following problem $\min_x \|Ax - b\|_2$ defined in Definition 1.2. They show that one step of gradient descent carries out data transformation as follows:

$$\|A(x + \delta_x) - b\|_2 = \|Ax - (b - \delta_b)\|_2$$
$$= \|Ax - \widetilde{b}\|_2$$

where $\delta_x$ denotes the one-step gradient descent on $x$ and $\delta_b$ denotes the corresponding data transformation on $b$. They also show that a self-attention layer is in principle capable of exploiting statistics in the current training data samples. Concretely, let $Q, K, V \in \mathbb{R}^{d \times d}$ denotes the weights for the query matrix, key matrix, and value matrix respectively. The linear self-attention layer updates an input sample by doing the following data transformation:

$$\widehat{b}_j = b_j + PVK^\top Q_j$$

where $\widehat{b}$ denotes the updated $b$ and $P$ denotes the projection matrix such that a Transformer step $\widehat{b}_j$ on every $j$ is identical to the gradient-induced dynamics $\widetilde{b}_j$. This equivalence implies that when training linear-self-attention-only Transformers for fundamental regression tasks, the models learned by GD and Transformers show great similarity.

[XRLM21] explores the occurrence of in-context learning during pre-training when documents exhibit long-range coherence. The Language Model (LLM) develops the ability to generate coherent next tokens by deducing a latent document-level concept. During testing, in-context learning is observed when the LLM deduces a shared latent concept between examples in a prompt. They demonstrate that in-context learning happens even when there is a distribution mismatch between prompts and pretraining data, especially when the pretraining distribution is a mixture of Hidden

Markov Models [BP66]. Theoretically, they show that the error of the in-context predictor is optimal when a distinguishability condition holds. In cases where this condition does not hold, the expected error still reduces as the length of each example increases. This finding highlights the importance of both input and input-output mapping for in-context learning.

## A.2 Transformer Theory

The advancements of Transformers have been noteworthy, however, their learning mechanisms are not completely comprehensible yet. Although these models have performed remarkably well in structured and reasoning activities, our comprehension of their mathematical foundations lags significantly behind. Past research has indicated that the outstanding performance of Transformer-based models can be attributed to the information within their components, such as multi-head attention. Various studies [TDP19, VB19, HL19, Bel22, LLS+24a, XSL24] have presented empirical proof that these components carry a substantial amount of information, which can help resolve different probing tasks.

Recent research has investigated the potential of Transformers through both theoretical and experimental methods, including Turing completeness [BPG20], function approximation [YBR+20, CDW+21], formal language representation [BAG20, EGZ20, YPPN21], and abstract algebraic operation learning [ZBB+22]. Some of these studies have indicated that Transformers may act as universal approximators for sequence-to-sequence operations [YBR+20, KS23, Ano24a] and emulate Turing machines [PMB19, BPG20]. [LWD+23] demonstrate the existence of contextual sparsity in LLM, which can be accurately predicted. They exploit the sparsity to speed up LLM inference without degrading the performance from both a theoretical perspective and an empirical perspective. [DCL+21] proposed the Pixelated Butterfly model that uses a simple fixed sparsity pattern to speed up the training of Transformer. Other studies have focused on the expressiveness of attention within Transformers [DGV+18, VBC20, ZKV+20, EGKZ21, SZKS21, WCM21, LSSY24, LSS+24a] and differentially private attention mechanisms [GSYZ24, LSSZ24a]. Recently, modern Hopfield models [HYW+23, HLSL24, WHHL24, HCL+24, HWL24a, HCW+24] have introduced Hopfield layers as powerful alternatives for transformer attention, offering solid theoretical guarantees and strong empirical performance [XHH+24, WHL+24]. Additionally, the statistical and computational theory of transformer-based diffusion models, specifically Diffusion Transformers (DiTs), has been studied in depth [HWL+24b, Ano24b].

Furthermore, [ZPGA23] has demonstrated that moderately sized masked language models may effectively parse and recognize syntactic information that helps in the partial reconstruction of a parse tree. Inspired by the language grammar model studied by [ZPGA23], [DGS23] consider the tensor cycle rank approximation problem. [GMS23] consider the exponential regression in neural tangent kernel over-parameterization setting. [LSZ23] studied the computation of regularized version of the exponential regression problem but they ignore the normalization factor. [DLS23] consider the softmax regression which considers the normalization factor compared to exponential regression problems [GMS23, LSZ23]. The majority of LLMs can perform attention computations in an approximate manner during inference, as long as there are sufficient guarantees of precision. This perspective has been studied by various research, including [CGRS19, KKL20, WLK+20, DKOD20, KVPF20, CDW+21, CDL+22, LLSS24b, LLS+24d, LSS+24b, SMN+24, LLS+24c, SSZ+24b, SSZ+24a]. With this in mind, [ZHDK23, AS23, BSZ24, DMS23, HYW+23, LLS+24b, CLS+24, LSSZ24b, HLSL24, HWL+24b, HWL24a] have studied the attention matrix computation from the hardness perspective and developed faster algorithms.

# B Lipschitz with respect to $x$

In Section B.1, we give the preliminary to compute the Lipschitz. In Section B.2, we compute the Lipschitz of function $\exp(Ax)$ with respect to $x$. In Section B.3, we compute the Lipschitz of the function $\alpha$ with respect to $x$. In Section B.4, we compute the Lipschitz of function $\alpha^{-1}$ with respect to $x$.

## B.1 Preliminary

We can compute

$$\frac{\mathrm{d}L}{\mathrm{d}x} = g(x)$$

Let $\eta > 0$ denote the learning rate.

We update

$$x_{t+1} = x_t + \eta \cdot g(x_t)$$

**Definition B.1.** *We define $\delta_b \in \mathbb{R}^n$ to be the vector that satisfies the following conditions*

$$\|\langle \exp(Ax_{t+1}), \mathbf{1}_n \rangle^{-1} \exp(Ax_{t+1}) - b\|_2^2 = \|\langle \exp(Ax_t), \mathbf{1}_n \rangle^{-1} \exp(Ax_t) - b + \delta_b\|_2^2$$

Let $\{-1, +1\}^n$ denote a vector that each entry can be either $-1$ or $+1$. In the worst case, there are $2^n$ possible solutions, e.g.,

$$(\langle \exp(Ax_{t+1}), \mathbf{1}_n \rangle^{-1} \exp(Ax_{t+1}) - \langle \exp(Ax_t), \mathbf{1}_n \rangle^{-1} \exp(Ax_t)) \circ \{-1, +1\}^n$$

The norm of all the choices are the same. Thus, it is sufficient to only consider one solution as follows.

**Claim B.2.** *We can write $\delta_b$ as follows*

$$\delta_b = \underbrace{\langle \exp(Ax_{t+1}), \mathbf{1}_n \rangle^{-1} \exp(Ax_{t+1})}_{f(x_{t+1})} - \underbrace{\langle \exp(Ax_t), \mathbf{1}_n \rangle^{-1} \exp(Ax_t)}_{f(x_t)}.$$

*Proof.* The proof directly follows from Definition B.1. $\qquad\square$

For convenience, we split $\delta_b$ into two terms, and provide the following definitions

**Definition B.3.** *We define*

$$\delta_{b,1} := (\langle \exp(Ax_{t+1}), \mathbf{1}_n \rangle^{-1} - \langle \exp(Ax_t), \mathbf{1}_n \rangle^{-1}) \cdot \exp(Ax_{t+1})$$

$$\delta_{b,2} := \langle \exp(Ax_t), \mathbf{1}_n \rangle^{-1} \cdot (\exp(Ax_{t+1}) - \exp(Ax_t))$$

Thus, we have

**Lemma B.4.** *We have*

- 

$$\delta_b = \delta_{b,1} + \delta_{b,2}$$

- *We can rewrite $\delta_{b,1}$ as follows*

$$\delta_{b,1} = (\alpha(x_{t+1})^{-1} - \alpha(x_t)^{-1}) \cdot \exp(Ax_{t+1}),$$

- *We can rewrite $\delta_{b,2}$ as follows*

$$\delta_{b,2} = \alpha(x_t)^{-1} \cdot (\exp(Ax_{t+1}) - \exp(Ax_t)).$$

*Proof.* We have

$$\delta_b = \delta_{b,1} + \delta_{b,2}$$
$$= \alpha(x_{t+1})^{-1} \exp(Ax_{t+1}) - \alpha(x_t)^{-1} \exp(Ax_{t+1}) +$$
$$\quad \alpha(x_t)^{-1} \exp(Ax_{t+1}) - \alpha(x_t)^{-1} \exp(Ax_t)$$
$$= \alpha(x_{t+1})^{-1} \exp(Ax_{t+1}) - \alpha(x_t)^{-1} \exp(Ax_t)$$
$$= \langle \exp(Ax_{t+1}), \mathbf{1}_n \rangle^{-1} \exp(Ax_{t+1}) - \langle \exp(Ax_t), \mathbf{1}_n \rangle^{-1} \exp(Ax_t),$$

where the 1st step follows from the definitions of $\delta_b$, the 2nd step follows from the definitions of $\delta_{b,1}$ and $\delta_{b,2}$, the 3rd step follows from simple algebra, the 4th step comes from the definition of $\alpha$. $\quad\square$

## B.2 Lipschitz for function $\exp(Ax)$ with respect to $x$

**Lemma B.5.** *If the following conditions holds*

- *Let $A \in \mathbb{R}^{n \times d}$*
- *Let $\|A(y - x)\|_\infty < 0.01$*
- *Let $\|A\| \leq R$*
- *Let $x, y$ satisfy that $\|x\|_2 \leq R$ and $\|y\|_2 \leq R$*

*Then we have*

$$\| \exp(Ax) - \exp(Ay) \|_2 \leq 2\sqrt{n} R \exp(R^2) \cdot \|x - y\|_2.$$

*Proof.* We have

$$
\begin{aligned}
\| \exp(Ax) - \exp(Ay) \|_2 &\leq \| \exp(Ax) \|_2 \cdot 2 \|A(x - y)\|_\infty \\
&\leq \sqrt{n} \cdot \exp(\|Ax\|_2) \cdot 2 \|A(x - y)\|_\infty \\
&\leq \sqrt{n} \exp(R^2) \cdot 2 \|A(x - y)\|_2 \\
&\leq \sqrt{n} \exp(R^2) \cdot 2 \|A\| \cdot \|x - y\|_2 \\
&\leq 2\sqrt{n} R \exp(R^2) \cdot \|x - y\|_2
\end{aligned}
$$

where the 1st step follows from $\|A(y - x)\|_\infty < 0.01$ and Fact 2.1, the 2nd step comes from Fact 2.1, the 3rd step follows from Fact 2.2, the 4th step follows from Fact 2.2, the last step follows from $\|A\| \leq R$. $\qquad\square$

## B.3 Lipschitz for function $\alpha(x)$ with respect to $x$

We state a tool from previous work [DLS23].

**Lemma B.6** (Lemma 7.2 in [DLS23]). *If the following conditions hold*

- *Let $\alpha(x)$ be defined as Definition 3.3*

*Then we have*

$$|\alpha(x) - \alpha(y)| \leq \| \exp(Ax) - \exp(Ay) \|_2 \cdot \sqrt{n}.$$

## B.4 Lipschitz for function $\alpha(x)^{-1}$ with respect to $x$

We state a tool from previous work [DLS23].

**Lemma B.7** (Lemma 7.2 in [DLS23]). *If the following conditions hold*

- *Let $\langle \exp(Ax), \mathbf{1}_n \rangle \geq \beta$*
- *Let $\langle \exp(Ay), \mathbf{1}_n \rangle \geq \beta$*

*Then, we have*

$$|\alpha(x)^{-1} - \alpha(y)^{-1}| \leq \beta^{-2} \cdot |\alpha(x) - \alpha(y)|.$$

## C Softmax Function with respect to $A$

In this section, we consider the function with respect to $A$. We define function softmax $f$ as follows

**Definition C.1** (Function $f$, Reparameterized $x$ by $A$ in Definition 3.1). *Given a matrix $A \in \mathbb{R}^{n \times d}$. Let $\mathbf{1}_n$ denote a length-$n$ vector that all entries are ones. We define prediction function $f : \mathbb{R}^{n \times d} \to \mathbb{R}^n$ as follows*

$$f(A) := \langle \exp(Ax), \mathbf{1}_n \rangle^{-1} \cdot \exp(Ax).$$

Similarly, we reparameterized $x$ by $A$ for our loss function $L$. We define loss function $L$ as follows

**Definition C.2** (Loss function $L_{\exp}$, Reparameterized $x$ by $A$ in Definition 3.2). *Given a matrix $A \in \mathbb{R}^{n \times d}$ and a vector $b \in \mathbb{R}^{n \times d}$. We define loss function $L_{\exp} : \mathbb{R}^{n \times d} \to \mathbb{R}$ as follows*

$$L_{\exp}(A) := 0.5 \cdot \|\langle \exp(Ax), \mathbf{1}_n \rangle^{-1} \exp(Ax) - b\|_2^2.$$

For convenience, we define two helpful notations $\alpha$ and $c$ with respect to $A$ as follows:

**Definition C.3** (Normalized coefficients, Reparameterized $x$ by $A$ in Definition 3.3). *We define $\alpha : \mathbb{R}^{n \times d} \to \mathbb{R}$ as follows*

$$\alpha(A) := \langle \exp(Ax), \mathbf{1}_n \rangle.$$

*Then, we can rewrite $f(A)$ (see Definition C.1) and $L_{\exp}(A)$ (see Definition C.2) as follows*

- $f(A) = \alpha(A)^{-1} \cdot \exp(Ax)$.

- $L_{\exp}(A) = 0.5 \cdot \|\alpha(A)^{-1} \cdot \exp(Ax) - b\|_2^2$.

- $L_{\exp}(A) = 0.5 \cdot \|f(A) - b\|_2^2$.

**Definition C.4** (Reparameterized $x$ by $A$ in Definition 3.4). *We define function $c : \mathbb{R}^{n \times d} \in \mathbb{R}^n$ as follows*

$$c(A) := f(A) - b.$$

*Then we can rewrite $L_{\exp}(A)$ (see Definition C.2) as follows*

- $L_{\exp}(A) = 0.5 \cdot \|c(A)\|_2^2$.

# D    Lipschitz with respect to $A$

In Section D.1, we give the preliminary to compute the Lipschitz. In Section D.2, we show the upper bound of $\delta_b$ with respect to $A$. In Section D.3, we compute the Lipschitz of function $\exp(Ax)$ with respect to $A$. In Section D.4, we compute the Lipschitz of the function $\alpha$ with respect to $A$. In Section D.5, we compute the Lipschitz of function $\alpha^{-1}$ with respect to $A$.

## D.1    Preliminary

We define $\delta_b$ as follows

**Definition D.1** (Reparameterized $x$ by $A$ in Definition B.1). *We define $\delta_b \in \mathbb{R}^n$ to be the vector that satisfies the following conditions*

$$\|\langle \exp(A_{t+1}x), \mathbf{1}_n \rangle^{-1} \exp(A_{t+1}x) - b\|_2^2 = \|\langle \exp(A_t x), \mathbf{1}_n \rangle^{-1} \exp(A_t x) - b + \delta_b\|_2^2$$

**Claim D.2** (Reparameterized $x$ by $A$ in Definition B.2). *We can write $\delta_b$ as follows*

$$\delta_b = \underbrace{\langle \exp(A_{t+1}x), \mathbf{1}_n \rangle^{-1} \exp(A_{t+1}x)}_{f(A_{t+1})} - \underbrace{\langle \exp(A_t x), \mathbf{1}_n \rangle^{-1} \exp(A_t x)}_{f(A_t)}.$$

*Proof.* The proof directly follows from Definition D.1. $\qquad\square$

For convenient, we split $\delta_b$ into two terms, and provide the following definitions

**Definition D.3** (Reparameterized $x$ by $A$ in Definition B.3). *We define*

$$\delta_{b,1} := (\langle \exp(A_{t+1}x), \mathbf{1}_n \rangle^{-1} - \langle \exp(A_t x), \mathbf{1}_n \rangle^{-1}) \cdot \exp(A_{t+1}x)$$
$$\delta_{b,2} := \langle \exp(A_t x), \mathbf{1}_n \rangle^{-1} \cdot (\exp(A_{t+1}x) - \exp(A_t x))$$

Thus, we have

**Lemma D.4** (Reparameterized $x$ by $A$ in Lemma B.4). *We have*

- *We can rewrite $\delta_b \in \mathbb{R}^n$ as follows*

$$\delta_b = \delta_{b,1} + \delta_{b,2}$$

- *We can rewrite $\delta_{b,1} \in \mathbb{R}^n$ as follows*

$$\delta_{b,1} = (\alpha(A_{t+1})^{-1} - \alpha(A_t)^{-1}) \cdot \exp(A_{t+1}x),$$

- *We can rewrite $\delta_{b,2} \in \mathbb{R}^n$ as follows*

$$\delta_{b,2} = \alpha(A_t)^{-1} \cdot (\exp(A_{t+1}x) - \exp(A_t x)).$$

*Proof.* We have

$$
\begin{aligned}
\delta_b &= \delta_{b,1} + \delta_{b,2} \\
&= \alpha(A_{t+1})^{-1}\exp(A_{t+1}x) - \alpha(A_t)^{-1}\exp(A_{t+1}x) + \\
&\quad \alpha(A_t)^{-1}\exp(A_{t+1}x) - \alpha(A_t)^{-1}\exp(A_t x) \\
&= \alpha(A_{t+1})^{-1}\exp(A_{t+1}x) - \alpha(A_t)^{-1}\exp(A_t x) \\
&= \langle\exp(A_{t+1}x), \mathbf{1}_n\rangle^{-1}\exp(A_{t+1}x) - \langle\exp(A_t x), \mathbf{1}_n\rangle^{-1}\exp(A_t x),
\end{aligned}
$$

where the 1st step follows from the definitions of $\delta_b$, the 2nd step follows from the definitions of $\delta_{b,1}$ and $\delta_{b,2}$, the 3rd step comes from simple algebra, the 4th step comes from the definition of $\alpha$. $\square$

## D.2 Upper Bounding $\delta_b$ with respect to $A$

We can show that

**Lemma D.5** (Reparameterized $x$ by $A$ in Lemma 4.1)**.** *If the following conditions hold*

- *Let $\beta \in (0,1)$.*
- *Let $\delta_{b,1} \in \mathbb{R}^n$ be defined as Definition D.3.*
- *Let $\delta_{b,2} \in \mathbb{R}^n$ be defined as Definition D.3.*
- *Let $\delta_b = \delta_{b,1} + \delta_{b,2}$.*
- *Let $R \geq 4$.*

*We have*

- *Part 1.*

$$\|\delta_{b,1}\|_2 \leq 2\beta^{-2}n^{1.5}\exp(2R^2) \cdot \|A_{t+1} - A_t\|_2$$

- *Part 2.*

$$\|\delta_{b,2}\|_2 \leq 2\beta^{-1}\sqrt{n}R\exp(R^2) \cdot \|A_{t+1} - A_t\|_2$$

- *Part 3.*

$$\| \underbrace{f(A_{t+1}) - f(A_t)}_{\delta_b} \|_2 \leq 4\beta^{-2}n^{1.5}R\exp(2R^2) \cdot \|A_{t+1} - A_t\|_2$$

*Proof.* **Proof of Part 1.** We have

$$
\begin{aligned}
\|\delta_{b,1}\|_2 &\leq |\alpha(A_{t+1})^{-1} - \alpha(A_t)^{-1}| \cdot \|\exp(A_{t+1}x)\|_2 \\
&\leq |\alpha(A_{t+1})^{-1} - \alpha(A_t)^{-1}| \cdot \sqrt{n} \cdot \exp(R^2) \\
&\leq \beta^{-2} \cdot |\alpha(A_{t+1}) - \alpha(A_t)| \cdot \sqrt{n} \cdot \exp(R^2) \\
&\leq \beta^{-2} \cdot \sqrt{n} \cdot \|\exp(A_{t+1}x) - \exp(A_t x)\|_2 \cdot \sqrt{n} \cdot \exp(R^2) \\
&\leq \beta^{-2} \cdot \sqrt{n} \cdot 2\sqrt{n}R\exp(R^2)\|A_{t+1} - A_t\| \cdot \sqrt{n} \cdot \exp(R^2)
\end{aligned}
$$

$$= 2\beta^{-2}n^{1.5}R\exp(2R^2) \cdot \|A_{t+1} - A_t\|$$

where the first step follows from definition, the second step follows from assumption on $A$ and $x$, the third step follows Lemma D.8, the forth step follows from Lemma D.7, the fifth step follows from Lemma D.6.

**Proof of Part 2.**

We have

$$\begin{aligned}
\|\delta_{b,2}\|_2 &\leq |\alpha(A_{t+1})^{-1}| \cdot \|\exp(A_{t+1}x) - \exp(A_t x)\|_2 \\
&\leq \beta^{-1} \cdot \|\exp(A_{t+1}x) - \exp(A_t x)\|_2 \\
&\leq \beta^{-1} \cdot 2\sqrt{n}R\exp(2R^2) \cdot \|A_{t+1} - A_t\|
\end{aligned}$$

**Proof of Part 3.**

We have

$$\begin{aligned}
\|\delta_b\|_2 &= \|\delta_{b,1} + \delta_{b,2}\|_2 \\
&\leq \|\delta_{b,1}\|_2 + \|\delta_{b,2}\|_2 \\
&\leq 2\beta^{-2}n^{1.5}R\exp(2R^2) \cdot \|A_{t+1} - A_t\| + 2\beta^{-1}n^{0.5}R\exp(2R^2) \cdot \|A_{t+1} - A_t\| \\
&\leq 2\beta^{-2}n^{1.5}R\exp(2R^2) \cdot \|A_{t+1} - A_t\| + 2\beta^{-2}n^{1.5}R\exp(2R^2) \cdot \|A_{t+1} - A_t\| \\
&\leq 4\beta^{-2}n^{1.5}R\exp(2R^2) \cdot \|A_{t+1} - A_t\|
\end{aligned}$$

where the 1st step follows from the definition of $\delta_b$, the 2nd step comes from triangle inequality, the 3rd step comes from the results in Part 1 and Part 2, the 4th step follows from the fact that $n \geq 1$ and $\beta^{-1} \geq 1$, the 5th step follows from simple algebra. $\qquad\square$

### D.3 Lipschitz for function $\exp(Ax)$ with respect to $A$

**Lemma D.6** (Reparameterized $x$ by $A$ in Lemma B.5). *If the following conditions holds*

- *Let $A, B \in \mathbb{R}^{n \times d}$*
- *Let $\|(A - B)x\|_\infty < 0.01$*
- *Let $\|A\| \leq R$*
- *Let $x$ satisfy that $\|x\|_2 \leq R$*

*Then we have*

$$\|\exp(Ax) - \exp(Bx)\|_2 \leq 2\sqrt{n}R\exp(R^2) \cdot \|A - B\|.$$

*Proof.* We have

$$\begin{aligned}
\|\exp(Ax) - \exp(Bx)\|_2 &\leq \|\exp(Ax)\|_2 \cdot 2\|(A - B)x\|_\infty \\
&\leq \sqrt{n} \cdot \exp(\|Ax\|_2) \cdot 2\|(A - B)x\|_\infty \\
&\leq \sqrt{n}\exp(R^2) \cdot 2\|(A - B)x\|_2 \\
&\leq \sqrt{n}\exp(R^2) \cdot 2\|A - B\| \cdot \|x\|_2 \\
&\leq 2\sqrt{n}R\exp(R^2) \cdot \|A - B\|
\end{aligned}$$

where the 1st step follows from $\|A(y - x)\|_\infty < 0.01$ and Fact 2.1, the 2nd step follows from Fact 2.1, the 3rd step follows from Fact 2.2, the 4th step comes from Fact 2.2, the last step follows from $\|A\| \leq R$. $\qquad\square$

### D.4 Lipschitz for function $\alpha(A)$ with respect to $A$

**Lemma D.7** (Reparameterized $x$ by $A$ in Lemma B.6). *If the following conditions hold*

- *Let $\alpha(A)$ be defined as Definition C.3*

*Then we have*

$$|\alpha(A) - \alpha(B)| \le \| \exp(Ax) - \exp(Bx)\|_2 \cdot \sqrt{n}.$$

*Proof.* We have

$$|\alpha(A) - \alpha(B)| = |\langle \exp(Ax) - \exp(Bx), \mathbf{1}_n \rangle|$$
$$\le \| \exp(Ax) - \exp(Bx)\|_2 \cdot \sqrt{n}$$

where the 1st step comes from the definition of $\alpha(x)$, the 2nd step follows from Cauchy-Schwarz inequality (Fact 2.1). □

### D.5 Lipschitz for function $\alpha(A)^{-1}$ with respect to $A$

**Lemma D.8** (Reparameterized $x$ by $A$ in Lemma B.7). *If the following conditions hold*

- *Let $\langle \exp(Ax), \mathbf{1}_n \rangle \ge \beta$*
- *Let $\langle \exp(Bx), \mathbf{1}_n \rangle \ge \beta$*

*Then, we have*

$$|\alpha(A)^{-1} - \alpha(B)^{-1}| \le \beta^{-2} \cdot |\alpha(A) - \alpha(B)|.$$

*Proof.* We can show that

$$|\alpha(A)^{-1} - \alpha(B)^{-1}| = \alpha(A)^{-1} \alpha(B)^{-1} \cdot |\alpha(A) - \alpha(B)|$$
$$\le \beta^{-2} \cdot |\alpha(A) - \alpha(B)|$$

where the 1st step follows from simple algebra, the 2nd step follows from $\alpha(A) \ge \beta, \alpha(B) \ge \beta$. □

## E Experiments

In this section, we show the complete numerical experimental results supporting our theoretical results that when training self-attention-only Transformers for softmax regression tasks, the models learned by gradient-descent and Transformers show great similarity.

**Experiments setup.** According to Definition 1.3, we construct the synthetic softmax regression tasks consists of randomly sampled length-$n$ documents $A \in \mathbb{R}^{n \times d}$ where each word has the $d$-dimensional embedding and targets $b \in \mathbb{R}^n$. In our experiments we choose a set of different value of document length $n \in \{25, 50, 100, 200, 400\}$ and a set of different embedding size $d \in \{5, 10, 20, 35, 50\}$. Following [ONR$^+$22], we compare the two models in our experiment: a trained single self-attention (SA) layer with a softmax unit approximating the full Transformers, and a softmax regression model trained with one-step gradient descent. The training objective for both models is defined as in Definition 1.3. For the single self-attention layer with a softmax unit, we choose the learning rate $\eta_{\mathrm{SA}} = 0.005$. For the softmax regression model, we determine the optimal learning rate $\eta_{\mathrm{GD}}$ by minimizing the $\ell_2$ regression loss over a training set of $10^3$ tasks through line search.

To compare the trained single self-attention layer with a softmax unit and the softmax regression model trained with one-step gradient descent, we sample $10^3$ tasks and record the losses of two models. In addition, we follow [ONR$^+$22] to record

- **Pred Diff**: the predictions difference measured with the $\ell_2$ norm:

$$\|\widehat{y}_{\mathrm{SA}}(A) - \widehat{y}_{\mathrm{GD}}(x)\|_2$$

  where $\widehat{y}_{\mathrm{SA}}(A)$ is corresponding to the $\widetilde{b}$ in Theorem 4.2, and $\widehat{y}_{\mathrm{GD}}(x)$ is corresponding to the $\widetilde{b}$ in Theorem 4.3.
- **Model Cos**: the cosine similarity between the sensitivities of two models:

$$\mathtt{CosSim}\left(\frac{\partial \widehat{y}_{\mathrm{GD}}(x)}{\partial x}, \frac{\partial \widehat{y}_{\mathrm{SA}}(A)}{\partial A}\right)$$

- **Model Diff**: the model sensitivity difference measured with the $\ell_2$ norm:

$$\|\frac{\partial \widehat{y}_{\mathrm{GD}}(x)}{\partial x} - \frac{\partial \widehat{y}_{\mathrm{SA}}(A)}{\partial A}\|_2$$

All experiments run on a single NVIDIA RTX2080Ti GPU with 10 independent repetitions.

**Results on tasks of different document lengths.** The results of the comparisons between a trained single self-attention layer and one-step gradient descent on synthetic softmax regression tasks of document length $n \in \{25, 50, 100, 200, 400, 1000\}$ and word embedding size $d = 20$ are shown in Figure 4-9. We measure two models' losses and similarities over the training steps of the SA layer for each set of tasks. From the results, we observe identical performances of the two models measured in losses. We also observe considerable alignment of the two models across tasks of different document lengths, indicated by decreasing prediction and model difference and increasing cosine similarity between models.

**Results on tasks of different word embedding sizes.** The results of the comparisons between a trained single self-attention layer and one-step gradient descent on synthetic softmax regression tasks of document length $n = 200$ and word embedding size $d \in \{5, 10, 20, 35, 50\}$ are shown in Figure 7 and 10-13. Similarly, we measure two models' losses and similarities over training steps of the SA layer for each set of tasks. We again observe similar performances and close alignment of the two models.

In conclusion, our experimental results empirically validate our theoretical results in Section 4, showing that when training self-attention-only Transformers for softmax regression tasks, the models learned by gradient-descent and Transformers show great similarity. Due to the non-linearity of softmax regression, it is not expected for models to match exactly as implied in our theoretical results in Section 4, which is also observed in our experimental findings.

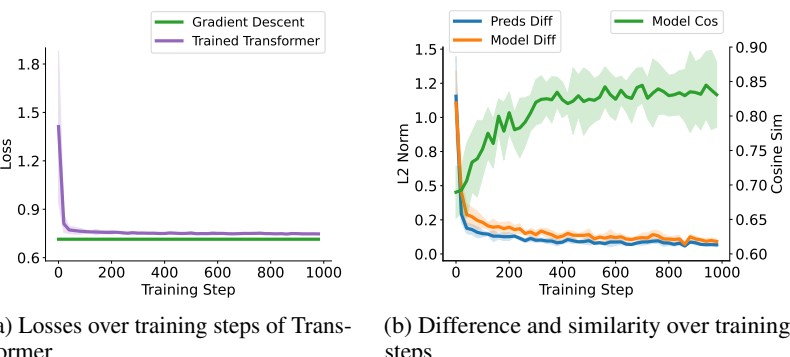

(a) Losses over training steps of Transformer

(b) Difference and similarity over training steps

Figure 4: Comparison between trained single-SA-layer Transformer and one-step GD on softmax regression tasks of document length $n = 25$ and embedding size $d = 20$.

# F   Limitations

Our findings are restricted to small Transformer and simple regression problems. One interesting direction for further investigation is to acquire a comprehensive perception of in-context learning in larger models. To our best knowledge, we believe this work does not have any negative societal impact.

# G   Impact Statements

This paper presents work whose goal is to advance the field of Machine Learning. There are many potential societal consequences of our work, none which we feel must be specifically highlighted here.

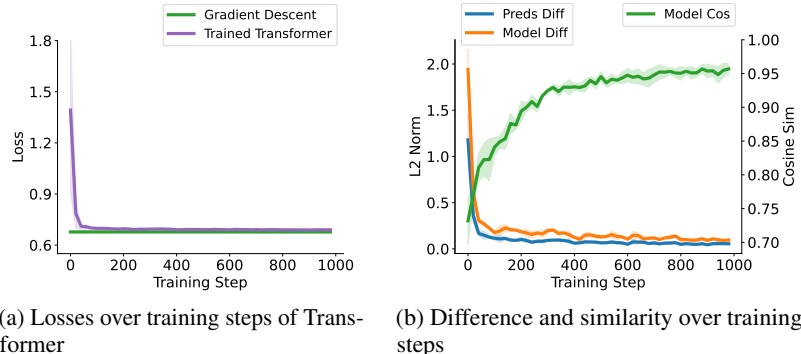

(a) Losses over training steps of Transformer

(b) Difference and similarity over training steps

Figure 5: Comparison between trained single-SA-layer Transformer and one-step GD on softmax regression tasks of document length $n = 50$ and embedding size $d = 20$.

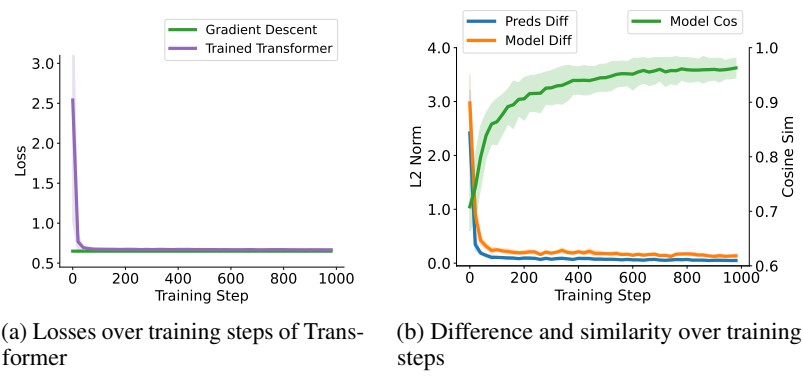

(a) Losses over training steps of Transformer

(b) Difference and similarity over training steps

Figure 6: Comparison between trained one-SA-layer Transformer and one-step GD on softmax regression tasks of document length $n = 100$ and embedding size $d = 20$.

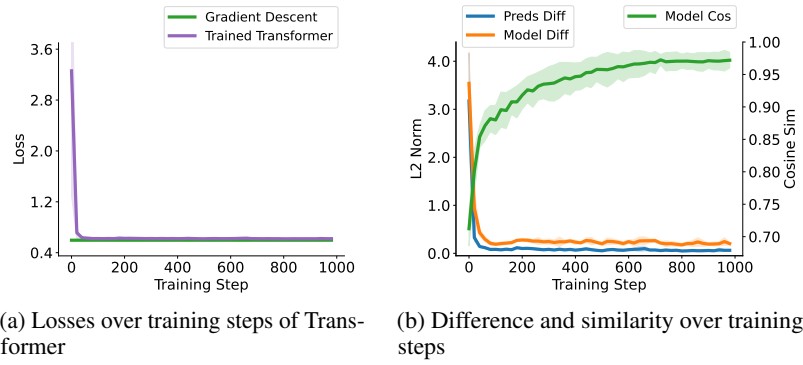

(a) Losses over training steps of Transformer

(b) Difference and similarity over training steps

Figure 7: Comparison between trained one-SA-layer Transformer and one-step GD on softmax regression tasks of document length $n = 200$ and embedding size $d = 20$.

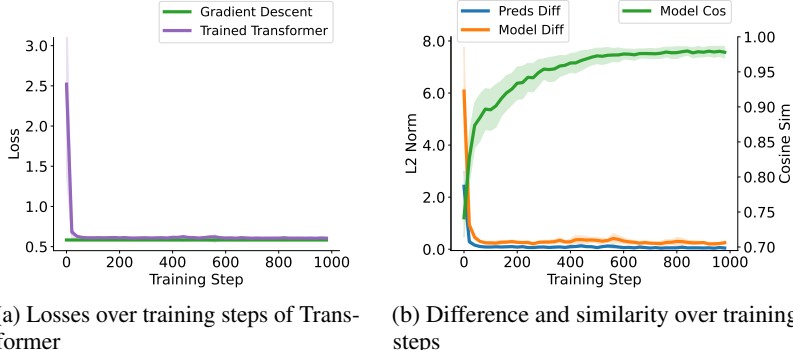

(a) Losses over training steps of Transformer

(b) Difference and similarity over training steps

Figure 8: Comparison between trained one-SA-layer Transformer and one-step GD on softmax regression tasks of document length $n = 400$ and embedding size $d = 20$.

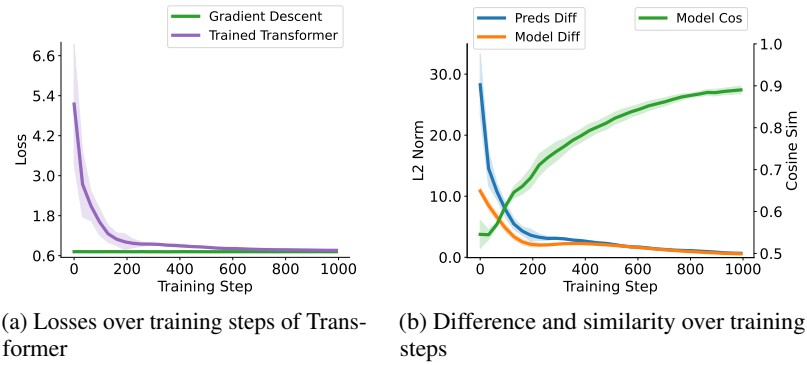

(a) Losses over training steps of Transformer

(b) Difference and similarity over training steps

Figure 9: Comparison between trained one-SA-layer Transformer and one-step GD on softmax regression tasks of document length $n = 1000$ and embedding size $d = 20$.

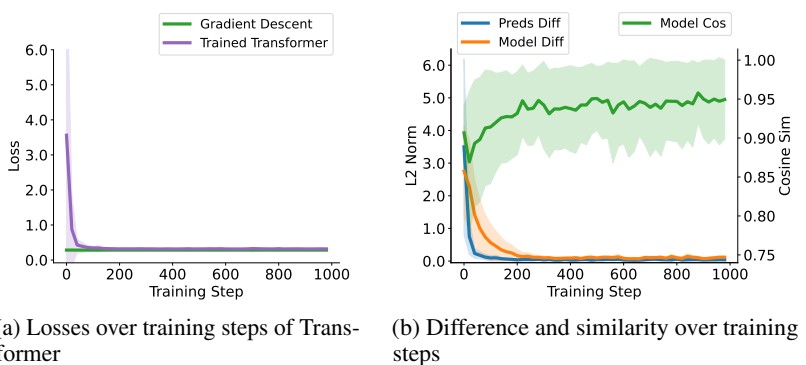

(a) Losses over training steps of Transformer

(b) Difference and similarity over training steps

Figure 10: Comparison between trained one-SA-layer Transformer and one-step GD on softmax regression tasks of document length $n = 200$ and embedding size $d = 5$.

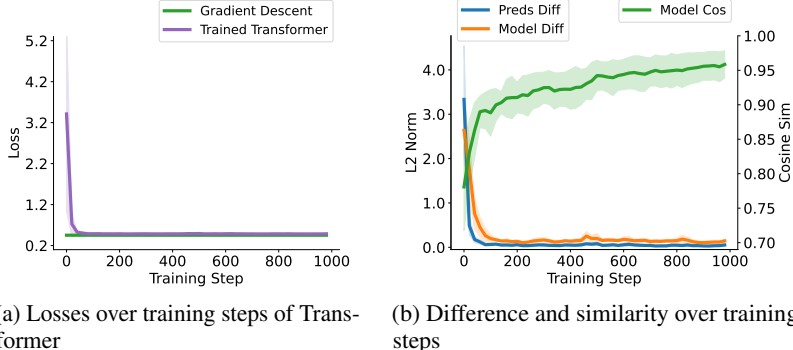

(a) Losses over training steps of Trans-
former

(b) Difference and similarity over training
steps

Figure 11: Comparison between trained one-SA-layer Transformer and one-step GD on softmax
regression tasks of document length $n = 200$ and embedding size $d = 10$.

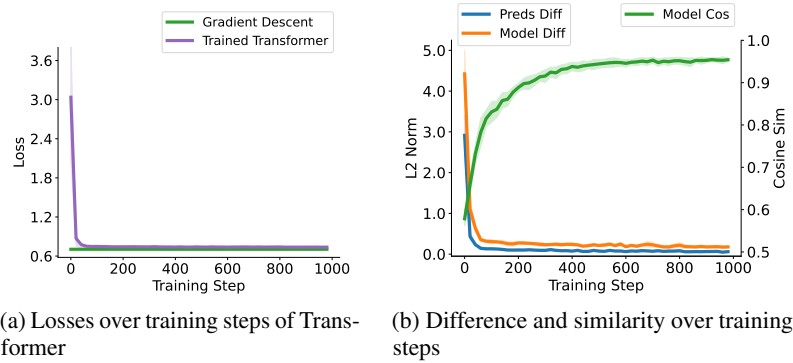

(a) Losses over training steps of Trans-
former

(b) Difference and similarity over training
steps

Figure 12: Comparison between trained one-SA-layer Transformer and one-step GD on softmax
regression tasks of document length $n = 200$ and embedding size $d = 35$.

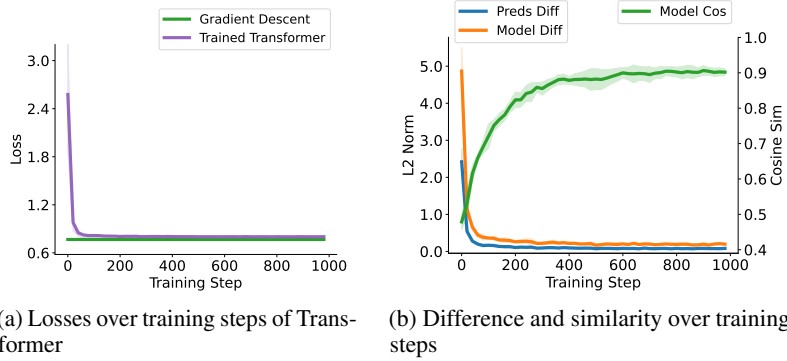

(a) Losses over training steps of Trans-
former

(b) Difference and similarity over training
steps

Figure 13: Comparison between trained one-SA-layer Transformer and one-step GD on softmax
regression tasks of document length $n = 200$ and embedding size $d = 50$.

