# OpenReview forum: "The Closeness of In-Context Learning and Weight Shifting for Softmax Regression"
_NeurIPS.cc/2024/Conference — NeurIPS 2024 poster_

### Official Review · Reviewer_K6fK · 2024-07-13

**Soundness:** 2
**Presentation:** 2
**Contribution:** 2
**Rating:** 5
**Confidence:** 2

**Summary:**

This paper aims to investigate why transformers possess the capability of in-context learning from a theoretical perspective. Previous works have shown simplified self-attention layer’s capability of learning linear functions in context. This work conducts further research based on softmax regression, as softmax-based algorithms are more complex and closer to algorithms used in actual LLMs. Through mathematical analysis and experimental validation, the authors conclude that the update acquired through gradient descent and in-context learning are similar when training simplified transformers for softmax regression tasks.

I am not fully follow this paper and I wish to learn more intuitive understanding of this paper during rebuttal (from the authors and other reviewers). I may adjust my rating score after rebuttal period.

**Strengths:**

1.Explaining the reasons why LLMs can learn from context at a theoretical level is of significant importance, and this paper has made a valuable exploration into this issue.

2.There is a thorough and comprehensive mathematical analysis to demonstrate the similarity between the models learned by transformers and gradient-descent on softmax regression.

3.Theoretical results and experimental results corroborate each other in this paper.

**Weaknesses:**

1.This paper appears to build upon previous research that explored in-context learning capabilities of transformer using linear regression, but instead opts for a softmax regression approach, lacking significant innovation and contribution.

2.Although this paper represents a further advancement over previous studies based on linear regression, it still employs a highly simplified Transformer model, which is insufficient to fully explain the principles of LLMs’ in-context learning ability.

3.I suggest that the structure of the paper could be organized more clearly, allowing readers to grasp the overall framework first. The details of the two models compared in the experiments should be explained more thoroughly. (1) The section 2 and 3 are not so well-organized, where some formula lacks necessary comments and explanations. (2) The introduction section is not so intuitive to follow.  (3) There is almost no textual descriptions about the proposed method section (section 3). I believe most NeurIPS reviewers are hard to follow the section 3 without enough descriptions.

**Questions:**

None

**Limitations:**

The limitations are fine with me.

---

> ### Author Rebuttal · Authors · 2024-08-07
>
> ***Q1: This paper appears to build upon previous research that explored in-context learning capabilities of transformer using linear regression, but instead opts for a softmax regression approach, lacking significant innovation and contribution.***
>
>
> Thank you for your comments. Different from previous works studying linear models with a linear self-attention layer, we take a step further to consider the important softmax unit for Transformers and provide analysis through softmax regression. Such extension approximating the non-linearity of Transformers is non-trivial and we believe our theoretical results provide novel insights for understanding the in-context learning capability of Transformers​.
>
>
> ***Q2: Although this paper represents a further advancement over previous studies based on linear regression, it still employs a highly simplified Transformer model, which is insufficient to fully explain the principles of LLMs’ in-context learning ability.***
>
>
> Thanks for your comments. We kindly argue that it is common to study shallow networks for theoretical analysis [1,2,3, 4]. While our analysis uses a simplified Transformer model, it serves as a foundational study, which can be extended to more complex Transformer architectures in future works and validated in more complex settings. We believe such study is crucial for understanding the fundamental mechanisms before expanding to more complex scenarios.
>
>
> [1] Nanda, Neel, et al. "Progress measures for grokking via mechanistic interpretability." ICLR2023.
>
>
> [2] Morwani, Depen, et al. "Feature emergence via margin maximization: case studies in algebraic tasks." ICLR2024.
>
>
> [3]Alman, Josh, and Zhao Song. "Fast attention requires bounded entries." NeurIPS 2024.
>
>
> [4] Mahankali, Arvind, Tatsunori B. Hashimoto, and Tengyu Ma. "One Step of Gradient Descent is Provably the Optimal In-Context Learner with One Layer of Linear Self-Attention". ICLR 2024.
>
>
> ***Q3: I suggest that the structure of the paper could be organized more clearly, allowing readers to grasp the overall framework first. The details of the two models compared in the experiments should be explained more thoroughly. (1) The section 2 and 3 are not so well-organized, where some formula lacks necessary comments and explanations. (2) The introduction section is not so intuitive to follow. (3) There is almost no textual descriptions about the proposed method section (section 3). I believe most NeurIPS reviewers are hard to follow the section 3 without enough descriptions.***
>
> Thank you for your feedback. In Section 2, we provide the notations and some basic algebra used in the proof of our main results. In Section 3, we formally define the network and the loss function, which are necessary to obtain our theoretical results. To help readers better understand our paper, we have added in the revised version more intuitive explanations in Sections 2 and 3 between the definitions, and moved some unused facts to the Appendix to make the paper more concise and easier to understand.
> Regarding the introduction, since our main findings are from the theoretical results, it is necessary to define the notions and concepts used in Theorem 1.4 and such necessity may make the introduction less intuitive to follow. To help readers better follow this section, we have included additional discussions to present our theoretical findings of similarity between the models learned by gradient descent and Transformers as well as the experimental results.

---

> > ### Comment · Reviewer_K6fK · 2024-08-10
> > **Thanks for responding my comments**
> >
> > Thank you for your rebuttal. I still suggest the authors should make the paper easy to follow. Considering other reviewer's comments, I tend to keep my score currently and am waiting for other reviewers' further ideas.

---

> > > ### Author Response · Authors · 2024-08-13
> > > **Reply to Reviewer K6fK**
> > >
> > > Thank you for your response and your recognition of our work. As you suggested, we have improved the paper presentation as follows, which will be reflected in the revised version of our paper:
> > > 1. For the introduction in Section 1, we have added more intuitive explanations for our studied in-context learning problem and its mathematical formulation.
> > > 2. We have added additional comments in Sections 2 and 3 between the definitions to help readers’ understanding, and moved some facts that are not used in the main body to the Appendix to ensure clarity.
> > > 3. For experiments part in Section 5, we have included additional details of the experimental setup from the Appendix and additional discussions of our experimental findings validating our theoretical results.

---

### Official Review · Reviewer_EkHX · 2024-07-13

**Soundness:** 3
**Presentation:** 3
**Contribution:** 3
**Rating:** 8
**Confidence:** 2

**Summary:**

This paper explores the relationship between in-context learning and weight shifting in the context of softmax regression for large language models. The authors delve into the mathematical aspects and study the in-context learning based on a softmax regression formulation. They present theoretical results that imply when training self-attention-only Transformers for fundamental regression tasks, the models learned by gradient descent and Transformers show great similarity. The paper shows the upper bounds of the data transformations induced by a single self-attention layer with a softmax unit and by gradient descent on a ℓ2 regression loss for softmax prediction function. The authors provide a formal theoretical result and numerical experiments to validate the theoretical results.

**Strengths:**

1. The paper presents a novel perspective on the in-context learning abilities of large language models (LLMs) by examining the softmax regression formulation. This approach is innovative as it bridges the gap between the attention mechanism's role in LLMs and the mathematical understanding of in-context learning. The authors' decision to study the bounded shifts in data transformations induced by a single self-attention layer with a softmax unit is a significant contribution to the field, offering a fresh angle on the problem that has not been extensively explored in prior work.

2. The paper is well-structured and clearly written.

3. The quality of the paper is high, as evidenced by the rigorous theoretical analysis and the comprehensive numerical experiments conducted to validate the theoretical findings.

**Weaknesses:**

1. While the paper compares the performance of a single self-attention layer with a softmax unit and a softmax regression model trained with one-step gradient descent, it would be informative to include comparisons with state-of-the-art models and methods.

2. The paper could benefit from a deeper exploration of the explainability and interpretability of the models in the context of softmax regression.

**Questions:**

Please refer to Weaknesses.

**Limitations:**

Please refer to Weaknesses.

---

> ### Author Rebuttal · Authors · 2024-08-07
>
> ***Q1: While the paper compares the performance of a single self-attention layer with a softmax unit and a softmax regression model trained with one-step gradient descent, it would be informative to include comparisons with state-of-the-art models and methods.***
>
> Thank you for your suggestion. Our current analysis focuses on establishing a theoretical foundation for understanding in-context learning of Transformers through softmax regression tasks. Therefore, similar to previous works [1, 2, 3] studying a single linear attention layer, we provide our analysis and results on a single self-attention layer with a softmax unit. Empirical comparisons with state-of-the-art models, as you suggest, would indeed provide more informative and interesting findings, which we leave as a future work.
>
> ***Q2: The paper could benefit from a deeper exploration of the explainability and interpretability of the models in the context of softmax regression.***
>
> Thank you for your suggestion. We will include additional discussion on the explainability and interpretability of models in our studied softmax regression setting.
>
> [1] Von Oswald, Johannes, et al. "Transformers Learn In-Context by Gradient Descent. ICML 2023".
>
> [2] Zhang, Ruiqi, et al. "Trained Transformers Learn Linear Models In-Context". Journal of Machine Learning Research 25.49 (2024): 1-55.
>
> [3] Mahankali, Arvind, Tatsunori B. Hashimoto, and Tengyu Ma. "One Step of Gradient Descent is Provably the Optimal In-Context Learner with One Layer of Linear Self-Attention". ICLR 2024.

---

### Official Review · Reviewer_LX6R · 2024-07-29

**Soundness:** 2
**Presentation:** 2
**Contribution:** 2
**Rating:** 5
**Confidence:** 3

**Summary:**

This research delves into enhancing the comprehension of in-context learning through theoretical analysis. It builds on prior studies showcasing how a single self-attention layer can learn gradient steps in linear regression contexts. The authors extend this concept to softmax regression and give the upper bounds of the data transformations driven by gradient descent for a single self-attention layer. Empirical results in the appendix validate these theoretical advancements.

**Strengths:**

- A comprehensive preliminaries and mathematical notations are defined properly
- Studying the relationship between in-context learning and gradient-decent is helpful for understanding current LLMs

**Weaknesses:**

- The presentation is hard to follow. It's better to organize the formulation in a question-driven format, and it's unclear why bounding the single step of data transformation relates to building a connection between in-context learning and softmax weight shift. Consider optimizing the presentation of certain mathematical proofs by relocating them to the appendix for conciseness, and leaving more place in the main paper for intuition illustration and experiment.
- As the theoretical analysis and experiment only consider a single attention layer, it's unclear whether other components in Transformers, such as MLP and Layer Norm, will affect the conclusion in the paper.

**Questions:**

See above.

**Limitations:**

The authors adequately addressed the limitations

---

> ### Author Rebuttal · Authors · 2024-08-07
>
> ***Q1: The presentation is hard to follow. It's better to organize the formulation in a question-driven format, and it's unclear why bounding the single step of data transformation relates to building a connection between in-context learning and softmax weight shift. Consider optimizing the presentation of certain mathematical proofs by relocating them to the appendix for conciseness, and leaving more place in the main paper for intuition illustration and experiment.***
>
> Thank you for your feedback. We agree that the presentation can be enhanced for better clarity. We have restructured the paper by keeping necessary theoretical formulation and key findings in the main paper while relocating unused facts for proofs to the appendix. Our studied connection between in-context learning and softmax weight shift builds on the previous study [1] showing matching weights between in-context learning of a single linear transformer layer and gradient descent. We will include more intuitive explanations and illustrations on this part to aid comprehension.
>
> ***Q2: As the theoretical analysis and experiment only consider a single attention layer, it's unclear whether other components in Transformers, such as MLP and Layer Norm, will affect the conclusion in the paper.***
>
> Thank you for your comments. Our current analysis focuses on a single attention layer with a softmax unit introducing non-linearity in full Transformers, which extends previous works [1, 2, 3] considering only a single linear attention layer. Similar to [1, 2, 3], we believe our current work establishes the basic fundamentals necessary for future studies introducing more complex Transformer components.
>
> [1] Von Oswald, Johannes, et al. "Transformers Learn In-Context by Gradient Descent. ICML 2023".
>
> [2] Zhang, Ruiqi, et al. "Trained Transformers Learn Linear Models In-Context". Journal of Machine Learning Research 25.49 (2024): 1-55.
>
> [3] Mahankali, Arvind, Tatsunori B. Hashimoto, and Tengyu Ma. "One Step of Gradient Descent is Provably the Optimal In-Context Learner with One Layer of Linear Self-Attention". ICLR 2024.

---

> > ### Comment · Reviewer_LX6R · 2024-08-13
> >
> > Thank you for your rebuttal. Given the complexity of understanding in-context learning in modern LLMs, which indeed poses a significant challenge and entails long-term commitment, this paper shows some advancement compared to the previous efforts (as listed in the rebuttal), notably from linear to softmax. I will adjust the score to favor acceptance. However, I still suggest the authors to improve the paper presentation.

---

> > > ### Author Response · Authors · 2024-08-13
> > > **Reply to Reviewer LX6R**
> > >
> > > Thank you for your response and your recognition of our contribution. As you suggested, we have improved the paper presentation as follows, which will be reflected in the revised version of our paper:
> > > 1. We have moved some facts in Section 2 that are not used in the main body to the Appendix to keep the main paper concise and easier to follow.
> > > 2. We have included more intuitive explanations in Section 1 and 2 on the data transformation formulation of in-context learning.
> > > 3. For experiments part in Section 5, we have included additional details of the experimental setup from the Appendix and additional discussions of our experimental findings validating our theoretical results.

---

### Decision · Program_Chairs · 2024-09-25

**Decision:**

Accept (poster)

**Comment:**

This paper provides a theoretical analysis of in-context learning in Transformers, focusing on the role of the softmax unit within self-attention layers. By examining the upper bounds of data transformations and comparing these with gradient descent on a regression loss, the authors offer insights into the similarities between models learned by Transformers and traditional methods. While the study is primarily theoretical, it fills a critical gap in understanding how softmax contributes to in-context learning. Based on the reviews, I recommend to accept the paper.